# Dynamic acoustic optimization of pulse tube refrigerators for rapid cooldown

Ryan Snodgrass ®[1,2] ✉, Vincent Kotsubo[1,2], Scott Backhaus[1,2] & Joel Ullom ®[1,2]

Pulse tube refrigerators are a critical enabling technology for many disciplines that require low temperatures. These refrigerators dominate the total power consumption of most modern cryostats, including those that reach millikelvin temperatures using additional cooling stages. In state-of-the-art commercial pulse tube refrigerators, the acoustic coupling between the driving compressor and the refrigerator is fixed and optimized for operation at base temperature. We show that this optimization is incorrect during the cooldown process, which results in wasted power consumption by the compressor and slow cooldown speed. After developing analytic expressions that demonstrate the need for acoustic tuning as a function of temperature, we dynamically optimize the acoustics of a commercial pulse tube refrigerator and show that the cooldown speed can be increased to 1.7 to 3.5 times the original value. Acoustic power measurements show that loss mechanism(s)—and not the capacity of the compressor—limit the maximum cooling available at high temperatures, suggesting that even faster cooldown speeds can be achieved in the future. This work has implications for the accessibility of cryogenic temperatures and the cadence of research in many disciplines such as quantum computing.

A wide variety of modern science is performed at temperatures <1 K. These low temperatures suppress electrical and thermal noise sources, and make unusual quantum-mechanical phenomena more accessible. State-of-the-art quantum processors[1,2], devices for quantum communication[3], searches for dark matter[4–6], and cameras to study the early universe[7] are all cooled to the millikelvin regime by dilution refrigerators, which must first be precooled to around 4 K. This precooling is almost always accomplished using low-frequency (order 1 Hz) pulse tube refrigerators (PTRs). A typical low-frequency PTR consumes roughly 10 kW of electrical power to provide about 1 W of cooling near 4 K.

Pulse tube refrigerators[8,9] are traveling-wave thermoacoustic systems. Heat is pumped by high-pressure helium gas that is expanded when nearest the cold end of the refrigerator's regenerator (absorbing heat) and compressed when nearest the warm end (depositing heat). A compressor and rotary valve cyclically compress the gas inside the PTR, producing acoustic power ($\dot{E}_2$), which is the time-averaged product of the pressure $p_1$ and volume flow rate $U_1$ phasors. The subscripts in $p_1$ and $U_1$ denote first-order thermoacoustic quantities, while the subscript in $\dot{E}_2$ shows that acoustic power is the product of two first-order quantities. Pulse tube refrigerators are unique in that they use an acoustic network (Fig. 1) to produce traveling-wave phasing between $p_1$ and $U_1$ without any cold moving parts; therefore, they require no maintenance and create relatively small vibrations at their cold ends. The thermodynamic core of a PTR is the regenerator, a component that stores and exchanges heat to build the relatively small temperature decrease of a single cycle into a much larger one over many repeated cycles. A thin-walled tube filled only with helium (which we call the buffer tube) transports $\dot{E}_2$ from the cold heat exchanger to the acoustic network while insulating the cold end from the warm end. This last component is often called a pulse tube, but we prefer to use the term buffer tube because it is unambiguous that we are referring to a component of the PTR and not the refrigerator itself.

[1]National Institute of Standards and Technology, Boulder, CO 80305, USA. [2]Department of Physics, University of Colorado Boulder, Boulder, CO 80309, USA.
✉ e-mail: ryan.snodgrass@nist.gov

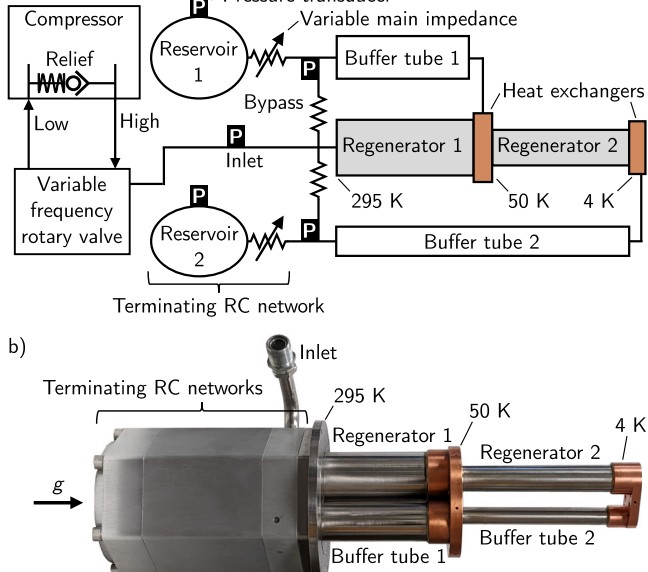

**Fig. 1 | A typical low-frequency, 4 K pulse tube refrigerator.** A schematic (**a**) and photograph (**b**) of the two-stage, dual-inlet PTR studied here, which is the same type that is typically used to precool dilution refrigerators. The variable main impedances and variable frequency rotary valve show the components that we tune dynamically, but that are traditionally static. The direction of gravity is shown by $g$.

Although they have been reliably and regularly used for many decades, PTRs still suffer from major inefficiencies. One issue—and the focus of this work—is that these refrigerators are optimized for performance only at their base temperature (usually near 4 K). However, during cooldown PTRs operate at all temperatures between ambient and base. At high temperatures, the parameters within the refrigerator's acoustic network are far from optimal, which leads to slow cooldown speeds and wasted compressor power input.

Cooldown speed dictates the cadence of low-temperature experiments. A dilution refrigerator used for basic research in quantum information science requires about a full day to cool to base temperature. Larger experiments with complete quantum processors, massive targets for rare event searches, or complex cameras can take many days to cool. An extreme example is the cryostat for the Cryogenic Underground Observatory for Rare Events (CUORE). The five PTRs in this tour de force of cryogenic engineering take about 20 days to precool the dilution refrigerator to 4 K[10]. Improving the cooldown speed of PTRs by factors of three or more (as we demonstrate here) provides the opportunity for increased measurement throughput and faster hardware modifications during the lifetime of such experiments.

The methods we demonstrate here may also be used to improve energy efficiency. In modern cryostats, PTRs are often oversized to achieve an acceptable cooldown speed, which results in far more cooling power being available at base temperature than is needed. These PTRs cost several thousand dollars a year to operate, and require special electrical and cooling infrastructure. If an improved PTR can cool three times faster at constant input power, then PTRs that consume three times less power can be substituted while preserving cooldown speed. Such a power reduction is highly desirable if ultralow temperatures and technologies are ever to become widely used.

The utility of dynamic optimization for PTRs originates in the temperature-dependent behavior of regenerators and buffer tubes. Assuming the regenerator has insignificant compliance and operates in the ideal-gas regime, it attenuates the volume flow rate that enters at the warm-end temperature $T_w$ by a factor of $1 - T_c/T_w$, where $T_c$ is the cold-end temperature (in the real-fluid regime, the thermal expansion

coefficient must be considered[11]). As we will soon show, this attenuation of $U_1$ in a cold PTR results in there being less flow available to generate $\dot{E}_2$. Additionally, the losses present in real buffer tubes impose heat loads on the cold heat exchanger that grow with $T_w - T_c$ and that are sensitive to the amplitude of $U_1$.

The terminating networks (Fig. 1) of PTRs largely control the amplitude and phase of $p_1$ and $U_1$ generated by the compressor, and thus provide an opportunity to dynamically tune the system. These *RC* networks are composed of a main impedance *R* (a needle valve) and a compliance *C* (a reservoir). The valve provides a real impedance that is sensitive to the position of the needle relative to its orifice, and the reservoir provides an imaginary impedance that is inversely proportional to the rotary valve frequency *f* and the reservoir volume *V*. The rotary valve decouples the drive frequency of the compressor from the frequency of oscillations in the PTR.

Single-temperature optimization of terminating networks (as is current practice) results in grossly unoptimized compressor-PTR coupling during cooldown. A compressor driving a commercial PTR at $T_c = 295$ K but optimized for $T_c = 4$ K typically shunts (wastes) a significant amount of its flow through a pressure relief valve because the impedances of the main needle valves are too high at that temperature. The pressure relief valve is installed in low-frequency PTR compressors to prevent overpressurization of the refrigerator before it cools; it essentially enables the refrigerator and compressor to be statically (although inefficiently) coupled as pressure and flow rate amplitudes change during cooldown. This inefficiency can be completely eliminated with dynamic acoustic optimization.

Previous efforts for the rapid cooldown of PTRs have been mostly limited to high-frequency coolers that operate at many 10s of Hz[12,13]. Those PTRs have fundamentally different acoustic networks from the type studied here, and therefore must use different methods for acoustic optimization. High-frequency PTRs are directly connected to linear compressors (no rotary valve required), and operate near resonance using inertance tubes or secondary displacers[14]. Inertance tubes are ineffective at shifting the phase between $p_1$ and $U_1$ in the low-frequency, Gifford-McMahon type PTRs studied here. While low-frequency PTRs are typically used in laboratory environments for precooling millikelvin refrigerators to about 4 K, high-frequency PTRs are used in aerospace applications and cannot usually cool from ambient temperature to 4 K.

The literature also contains one work concerning the cooldown speed of low-frequency PTRs[15]; however, this work studied a custom refrigerator that only cooled to 100 K, and the demonstrated improvement in cooldown speed was less than 10%. This work also failed to present an analysis that explains the need for optimization of acoustic settings as a function of temperature. No rapid cooldown studies have yet to consider 4 K PTRs, which is unfortunate because there is a large demand for these cryocoolers. They also span the highest temperature differences and therefore benefit the most from temperature-dependent optimization. We are unaware of any commercial PTR that utilizes dynamic acoustic optimization.

In this work, we present analytic expressions that guide the temperature-dependent optimization of low-frequency PTRs. We then report measurements from a common 4 K cooler (Cryomech PT407-RM, paired with the manufacturer-recommended CP2850 compressor). These measurements demonstrate that commercial PTRs can achieve cooldown speeds up to 3.5 times the default if their acoustics are optimized as a function of temperature and if a heat transfer mechanism is implemented between their two heat exchangers (so that cooling power increases from both stages may be utilized). We conclude the manuscript by showing that the cooling power of the PTR studied here is limited by loss mechanisms—likely in the buffer tube—and not by the power provided by the compressor. This suggests that normalized cooldown speeds greater than 3.5 can be achieved with no additional power input.

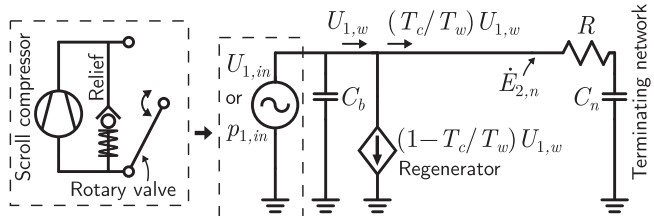

**Fig. 2 | Electrical circuit analogous to a low-frequency, single-stage orifice PTR.** The schematic of this refrigerator is the same as shown in Fig. 1a except without the second stage and with no bypass. The compressor and rotary valve system functions as a pressure or flow-rate source, depending on whether the relief valve is open or closed (this valve is installed to prevent overpressurization). The volume flow rate phasor at the regenerator's warm end is $U_{1,w}$.

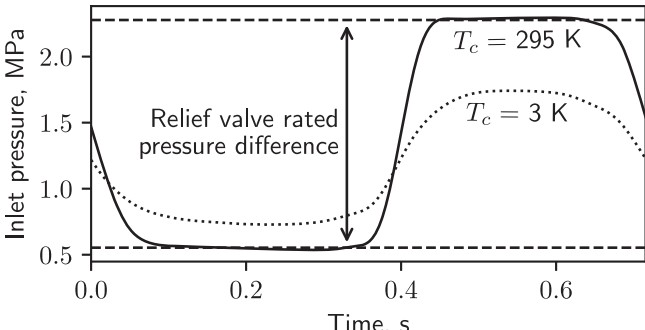

**Fig. 3 | Pressure amplitude limited by relief valve at ambient temperature.** The pressure oscillation measured at the inlet of the commercial PTR shown in Fig. 1b when the first and second stages were regulated near ambient temperature (solid line) and when no heat was applied to the cold ends in a steady state (dotted line). For both lines, the PTR was operated with the frequency and main orifice settings that produce optimal performance at base temperature.

## Results

### Basis for temperature-dependent optimization

In this section, we analyze how much acoustic power is delivered from the compressor to the PTR's terminating network ($\dot{E}_{2,n}$). This quantity is important because an ideal orifice pulse tube refrigerator (OPTR) has cooling power $\dot{Q}_c$ equal to $\dot{E}_{2,n}$. Although we are interested in PTRs with multiple stages and with acoustic networks that include bypasses (Fig. 1), we can analytically study the simplified geometry of a single-stage OPTR and reveal the fundamental principles that apply to more complicated systems. Furthermore, the majority of flow in dual-inlet[16] PTRs goes through the regenerators, not the bypasses. Our analysis shows that to maximize $\dot{E}_{2,n}$ the acoustic network must be tuned with temperature.

We proceed by simplifying an OPTR and its compressor into only the most essential features which determine $\dot{E}_{2,n}$. The appropriateness of these simplifications is confirmed by the qualitative agreement between the analytics and the following measurements. An electrical circuit driven by a sine wave is used in analogy to the PTR-compressor system (square waves are considered in Supp. Methods A). The analogs of pressure, volume flow rate, compliance, and acoustic resistance are (respectively) voltage, current, capacitance, and electrical resistance.

Low-frequency pulse tube refrigerators are driven by oil-lubricated scroll compressors operating at 50 or 60 Hz. These compressors deliver a volume flow rate that is approximately fixed by the geometry of two interleaving scrolls and their frequency of rotation. A rotary valve cyclically connects the PTR inlet to the high and low-pressure sides of the compressor at a frequency of a few Hz. Although we do not consider the details of this process here, it is inherently lossy because no work is recovered from the volume compressed each cycle[17,18]. The scroll compressor, its relief valve, and the rotary valve are shown in Fig. 2; taken together, these components provide the PTR with an alternating source of pressure or volume flow rate.

In Fig. 2 we also show the $RC$ terminating network of the OPTR, represented by resistance $R$ and compliance $C_n = V_n/\gamma p_m$, where $V_n$ is the volume of the reservoir, $\gamma$ is the ratio of the isobaric to isochoric specific heats, and $p_m$ is the mean pressure. The other compliance $C_b$ is a lumped parameter accounting for the compressible volume between the rotary valve and terminating network, including the buffer tube, regenerator void fraction, and (often) a helium hose between the valve and PTR.

The final component in the circuit is a sink representing the attenuation[11,19] of the volume flow rate in the regenerator as the temperature drops from warm $T_w$ to cold $T_c$. We consider a PTR with insignificant pressure drop across the regenerator—which our measurements show to be a good approximation (Fig. S1)—so the regenerator's hydrodynamic resistance is not included in the analogous circuit.

When a PTR optimized for 4 K is instead operated near ambient temperature (during cooldown, for example), its pressure amplitude is high and limited by the relief valve (Fig. 3). This causes a portion of the flow exiting the scroll compressor to be shunted between the high and low sides. The destruction of work availability across the relief valve makes it inherently undesirable to operate a PTR in this regime. With the relief valve open, the compressor and rotary valve system functions as a pressure source, and the pressure amplitude at the inlet $|p_{1,in}|$ is set at half the rating of the relief valve. The acoustic power in the network is

$$\dot{E}_{2,n} = \frac{(\omega R C_n)^2}{1+(\omega R C_n)^2}\frac{|p_{1,in}|^2}{2R}, \tag{1}$$

where $\omega = 2\pi f$. The above is derived in Supp. Methods B. If the real impedance of the terminating network is much larger than the imaginary impedance ($\omega R C_n \gg 1$), then $\dot{E}_{2,n}$ is approximately $|p_{1,in}|^2/2R$, and more power is received from the compressor only by decreasing $R$. This condition is met for the PTR studied here when its acoustics are optimized for low-temperature performance (see Fig. S2 for estimates of $R$ and $C_n$).

Since there is no temperature dependence in Eq. (1), $\dot{Q}_c$ should be nearly fixed in a cooling PTR with limited pressure amplitude. Figures S3 and S4 show measurements from the commercial PTR studied here, and confirm that when $p_{1,in}$ is limited by the relief valve $\dot{Q}_c$ at the second stage is insensitive to $T_c$. This same behavior does not hold at the first stage because of loss mechanisms that we discuss in a later section. We also achieved good quantitative agreement (Fig. S2) between $\dot{E}_{2,n}$ measured in the commercial cooler and a numerical simulation of the circuit in Fig. 2 using measured acoustic parameters (Fig. S5). To measure $\dot{E}_{2,n}$, pressure transducers were installed at a variety of locations (Fig. 1a) inside the PTR (see Supp. Methods C for details).

If the relief valve is closed then no work availability is destroyed across it, and the compressor and rotary valve system functions as a volume flow-rate source. The acoustic power in the network is

$$\dot{E}_{2,n} = \frac{R}{2}\frac{|U_{1,in}|^2}{(T_w/T_c + C_b/C_n)^2 + (\omega R C_b)^2}, \tag{2}$$

where $|U_{1,in}|$ is the amplitude of the volume flow rate exiting the rotary valve. The above is derived in Supp. Methods B. In this regime, $\dot{E}_{2,n}$ decreases as the PTR cools or as the compliance between rotary valve and terminating network increases: flow used to compress the volume before the network and flow consumed in the regenerator cannot generate $\dot{E}_{2,n}$.

Equation (2) forms the basis for acoustic impedance matching between compressor and PTR. Acoustic parameters such as $R$ may be changed to maximize the power delivered from the compressor as the PTR cools. Finding where $\partial \dot{E}_{2,n}/\partial R = 0$ gives the network resistance for maximizing $\dot{E}_{2,n}$:

$$R_{\max} = (T_w/T_c + C_b/C_n)/\omega C_b. \qquad (3)$$

The optimal main impedance increases as the pulse tube cools ($T_c$ decreases). This is the fundamental result that motivated our study of dynamic acoustic optimization for rapid cooldown. Measurements from a commercial PTR (Fig. S4) qualitatively agree with Eq. (2) and show that $\dot{E}_{2,n}$ at warm temperatures can be significantly increased by decreasing $R$ from its optimal value at base temperature.

In the following two sections, we show that markedly increased cooling powers can be obtained in a commercial PTR by decreasing $R$ near ambient temperature. However, by comparing $\dot{E}_{2,n}$ and $\dot{Q}_c$ measurements we know that significant loss mechanisms in this particular PTR cloud the intuition developed in this section: while trends in $\dot{E}_{2,n}$ match the predictions of Eqs. (1) and (2), $\dot{Q}_c$ only sometimes follows the same trends. In the last section of this manuscript we hypothesize on the loss mechanisms that cause this discrepancy.

## Available cooling with optimized acoustics

The acoustic networks of low-frequency, 4 K PTRs are usually the dual-inlet type[16]. This means that in addition to the main orifice placed immediately before the reservoir, there is a secondary orifice and flow path (the bypass) between the inlet and the warm end of the buffer tube (Fig. 1a). Relatively small changes to cooling power were observed when adjusting the second-stage bypass impedance, and therefore the bypass impedances were fixed for all results presented here.

Instead we turn our attention to the main needle valves. The manufacturer-installed needles were not designed to be repeatedly adjusted, so we replaced them with custom ones that had the same tip geometry. Stepper motors were installed exterior to the PTR and were used to change the position of the needles; encoders were used to track position. A stepper motor driver was also used to modulate the speed of the rotary valve, which determines the frequency $f$ of the pressure and flow rate oscillations inside the PTR.

In the remainder of the manuscript, we use the term cold impedance to refer to the main needle valve settings that produce optimal low-temperature performance (details in Fig. S6). When cold impedance is paired with the manufacturer-chosen $f$ (1.4 Hz), we use the term cold acoustics, which represents the performance of the PTR without dynamic acoustic optimization. We also present measurements with progressively decreased main impedance, which we call partly open and more open. Compared to cold impedance, the needles

in the partly open configuration were moved away from each orifice by 0.24 mm (stage 1) and 0.20 mm (stage 2), while the needles for the more open configuration were moved away by 1.12 mm (stage 1) and 1.32 mm (stage 2).

To measure the cooling power $\dot{Q}_c$ of the PTR at elevated temperatures, we mounted an array of resistive heaters to the first and second-stage heat exchangers, and then measured the heat required to regulate each stage to the same temperature. Figure 4a shows that with cold acoustics, the first stage provides 197 W of cooling ($\dot{Q}_{c,1}$) at 295 K, and the second stage is 63 W ($\dot{Q}_{c,2}$). The largest improvements to $\dot{Q}_c$ at 295 K are achieved when simultaneously increasing $f$ and opening the main needle valves (Fig. 4b, c). First and second-stage $\dot{Q}_c$ are increased to 1.7 and 1.9 times their cold-acoustic values by choosing the best combination of needle position and frequency. We did not test all combinations of these parameters, so even higher $\dot{Q}_c$ may be achievable. Cooling power at other temperatures is shown in Fig. S7.

It is important to note in Fig. 4 that the maximum cooling available at the first stage is 5.5 times that at the second stage with cold acoustics. For applications where the second stage cools more slowly than the first (for example, when a large experiment is attached to the second stage), it is very desirable to transfer heat between stages so that this large cooling power may be utilized. In the next section, we achieve this using gravity-driven heat pipes (thermosiphons)[20–22], although other methods may suffice[23].

The maximum available cooling normalized by the cold-acoustic value at the second stage is compiled as a function of temperature in Fig. 5. The results are separated by whether or not a heat transfer mechanism is implemented (so the cooling power of the first stage can be used at the second). At 295 K, $\dot{Q}_{c,2}$ can be increased to 7.3 times the original when optimizing the acoustics and utilizing the cooling power of both stages. Comparison to the data with heat transfer but without optimized acoustics (4.1 times the original at 295 K) shows that although heat exchange is important for increasing $\dot{Q}_{c,2}$, acoustic optimization is still required for the most significant improvements.

Figure 5 also shows that dynamic acoustic optimization provides the greatest improvements to $\dot{Q}_c$ at temperatures furthest from the base, where static optimization is traditionally performed. For the PTR studied here, the optimal main impedance is smallest at ambient temperature and increases towards the cold-impedance value as the temperature drops (Fig. S7)—the same trend predicted by Eq. (3). The optimal $f$ is highest at ambient temperature and decreases as temperature drops. This result is not predicted by Eq. (2) but is explained by loss mechanisms, which we discuss later.

## Cooldown speed comparison

The previous section showed that $\dot{Q}_c$ near ambient temperature can be significantly increased by making simple changes to a PTR's acoustic

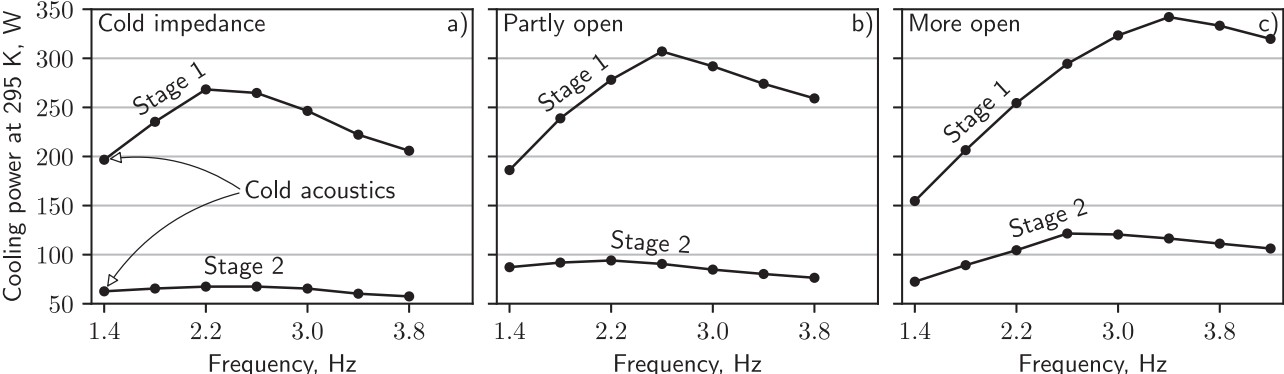

**Fig. 4 | Cooling power at ambient temperature for a variety of acoustic settings.** Subplot (**a**) gives cooling as a function of $f$ when the main needle valves were optimized for low-temperature performance. Subplots (**b**, **c**) give cooling when the first and second stage needle valves were progressively opened, reducing the acoustic impedance.

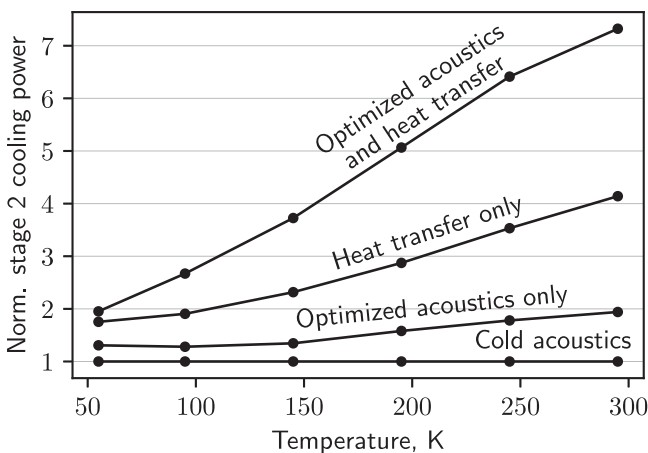

**Fig. 5 | Available cooling with optimized acoustics and/or heat transfer between stages.** Normalized cooling power at the second stage as a function of temperature. For lines with heat transfer, the cooling power was calculated assuming infinite thermal conductance between stages.

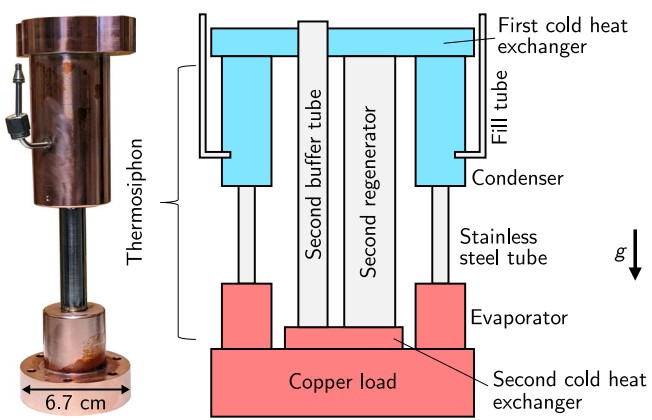

**Fig. 6 | Thermosiphons transfer heat between stages.** A photograph of one of the two manufactured thermosiphons, and a schematic of the cryostat between the first and second heat exchangers. The colors reflect that if a large mass is placed on the second stage, the first stage cools more quickly than the second.

parameters. Now we dynamically optimize a PTR during cooldown and measure the overall cooldown speed of the 4 K stage.

To transfer heat between the stages, we installed two thermosiphons (Fig. 6 and Supp. Methods D) between the first and second heat exchangers. With one filled with ethane and the other nitrogen, the thermosiphons provide high thermal conductance (approaching 10 W/K, Fig. S8) over most of the temperature range between 295 and 65 K (Fig. S9).

The cooling power of the first stage may only be utilized at the second if the first stage cools more quickly; this is the case for a cryostat with a relatively large mass on the 4 K stage, but is not the case for a bare PTR. We bolted 17.7 kg of copper to the second heat exchanger to emulate a large experimental load, such as a dilution refrigerator. In addition to the real mass, we sometimes added virtual mass by applying heat during cooldown (Supp. Methods E).

A cooldown (Fig. 7a) for a PTR set to cold acoustics and with the thermosiphons evacuated is compared to a cooldown for the same cryostat but with rapid cooldown improvements (dynamically optimized acoustics and the thermosiphons filled with ethane and nitrogen). With cold acoustics, the second stage cools in 39.7 h, while the rapid cooldown cryostat cools in 11.5 h (3.5 times the cooldown speed). For these cooldowns, the effective copper load was 71 kg, so when there was no heat transfer (evacuated thermosiphons), the second stage temperature greatly lagged the first, and the high cooling power of the first stage (Fig. 4) was not utilized. For the rapid cooldown configuration, the first stage cooled more slowly (even though $\dot{Q}_{c,1}$ was greatly increased) and tracked the temperature of the second stage, showing that the thermosiphons were effective and that the relatively large $\dot{Q}_{c,1}$ at high temperatures was used at the second stage.

When a heat transfer mechanism is installed in the cryostat, the benefit of dynamic acoustic optimization depends on the size of the loads at each stage. As a metric for the size of the loads, we compare the enthalpy to remove during cooldown from all metal mounted to the second stage $\Delta h_2$ to the first stage $\Delta h_1$. To study the effect of $\Delta h_2/\Delta h_1$, we varied the virtual mass on the second stage (Supp. Methods E) and measured the rapid cooldown speed, normalizing by the cold-acoustic speed for the same mass (Fig. 7b). When both thermosiphons were filled and the acoustics were optimized, the normalized cooldown speed was between 2.6 ($\Delta h_2/\Delta h_1 = 2.3$) and 3.5 ($\Delta h_2/\Delta h_1 = 8.6$). The shape of the curve suggests minimal improvement for $\Delta h_2/\Delta h_1 > 8.6$. A more extensive dataset under a variety of acoustic settings and thermosiphon fluids may be found in Fig. S12.

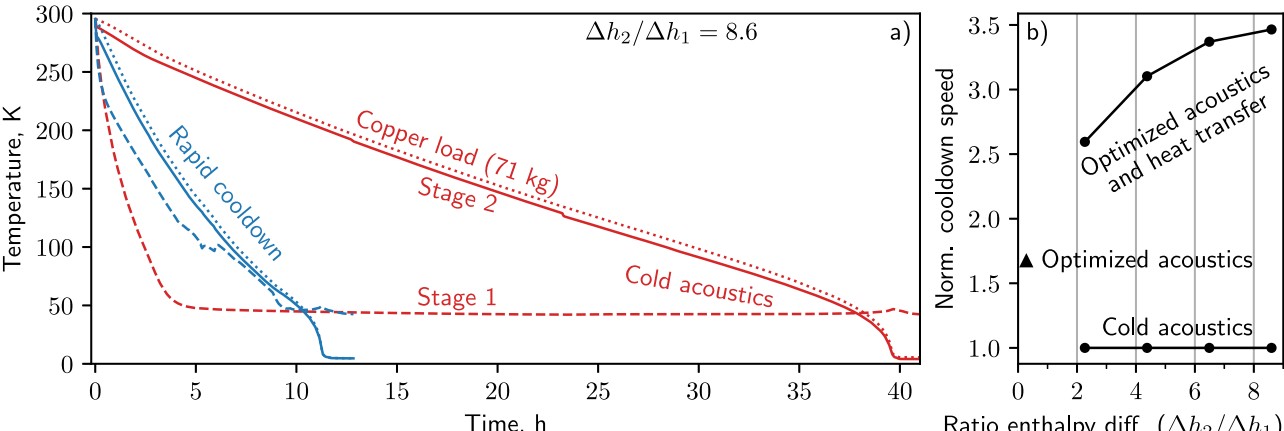

**Fig. 7 | Cooldowns with and without dynamic acoustic optimization.**
**a** Cooldown curves for the PTR when statically optimized for low temperature (red lines) and when dynamically optimized with heat transfer between stages (blue lines). Cooldown was considered complete once the second stage reached 6 K (final temperature was around 4 K or 5 K but varied between experiments, perhaps because the base temperature is very sensitive to needle position). **b** Normalized cooldown speed versus the ratio of the enthalpy to remove from each stage's load between 295 and 4 K. Figure S10 shows the needle positions and frequency schedule as a function of temperature. For the $\Delta h_2/\Delta h_1 = 0.3$ data point, the needle and frequency schedule was unique (Fig. S11).

Next we quantified the cooldown speed improvement at the second stage from dynamic acoustic optimization alone. To do this, the 17.7 kg copper load was removed, which broke the conduction path between the evaporators and the second heat exchanger (the thermosiphons were unable to transfer heat from the second stage to the first). No virtual heat was applied during this experiment; that, combined with the lack of copper load, resulted in $\Delta h_2/\Delta h_1 = 0.3$ (the leftmost data point in Fig. 7b). The cooldown speed here−1.7 times the cold-acoustic value−provides an estimate for any cryostat without heat transfer between stages.

We also note that the normalized cooldown speed is influenced by the thermal conductance $G$ between the heat exchangers and the 4 K load. For $G \to \infty$, there is no temperature difference between load and heat exchanger, which is optimal because $\dot{Q}_c$ drops with temperature (Fig. S3). For $G \to 0$, dynamic acoustic optimization provides no benefit because the cold stages reach their base temperature while the load is still near ambient temperature. At that point it is optimal for the PTR to revert to cold acoustics. We achieved a high thermal conductance (8.2 W/K at 295 K) between the copper load and second heat exchanger by pressing them together with a 0.2-mm thick layer of indium in between.

## Loss mechanisms

Temperature-dependent optimization of PTRs was previously motivated using Eqs. (2) and (3). While acoustic power measurements do qualitatively follow Eq. (2), cooling power measurements only sometimes follow the same trends. For example, Eq. (2) suggests that $\omega$ should be minimized at all temperatures to maximize $\dot{E}_{2,n}$, while Fig. 4 shows that higher $f = \omega/2\pi$ can sometimes increase $\dot{Q}_c$. In this section, we discuss loss mechanisms that may cause this discrepancy.

An ideal regenerator in the ideal-gas regime transports zero total power towards its cold end and an ideal buffer tube transports total power away from the cold heat exchanger equal to the acoustic power. Total power accounts for all power flows and is the quantity of interest when performing First Law analyses[19]. Therefore, the cooling power available in an ideal PTR is equal to the acoustic power flow through the buffer tube $\dot{E}_{2,bt}$. We do not have a direct measurement of $\dot{E}_{2,bt}$, but $\dot{E}_{2,n}$ should serve as a good estimate if the flow through the bypasses is small compared to the flow through the regenerators.

Figure 8a plots the first-stage network acoustic power $\dot{E}_{2,n1}$ and cooling power $\dot{Q}_{c,1}$ with the needles in their cold-impedance positions.

A critical observation is that lowering $f$ increases $\dot{E}_{2,n1}$ but does not always increase $\dot{Q}_{c,1}$. We hypothesize that a significant loss mechanism limits the cooling power available at low frequencies because otherwise, increases to $\dot{E}_{2,n1}$ should be accompanied by increases to $\dot{Q}_{c,1}$.

A likely source of loss is acoustic shorting of the buffer tube by helium that reaches both cold and warm heat exchangers during a single cycle. The helium in the buffer tube acts as a fluid piston, and when shorting occurs, this piston has a stroke $2|\xi_1|$ that is near or greater than the buffer tube length $L$, where the displacement amplitude $|\xi_1|$ is

$$|\xi_1| = \frac{|U_1|}{2\pi f A}, \qquad (4)$$

and the buffer tube's cross-sectional area is $A$. We will refer to this loss mechanism as large-stroke loss.

Figure 8a may suggest that large-stroke loss is only present below about 2.2 Hz. Below this frequency, $\dot{Q}_{c,1}$ drops significantly, while above it $\dot{Q}_{c,1}$ changes by an amount almost equal to $\dot{E}_{2,n1}$ increments (as expected in the absence of losses). The length of this buffer tube was likely chosen to avoid large-stroke loss when its cold end is near 50 K, where $|U_1|$ (and therefore $|\xi_1|$) is smaller, attenuated by the temperature gradient in the regenerator. At 295 K and with cold-acoustic parameters (i.e., under normal operating conditions during cooldown), the first-stage buffer tube is very lossy (by comparison to the 2.2 Hz data, the cooling power at 1.4 Hz would be 90 W higher if there were no losses). We estimate that the stroke in the first-stage buffer tube is $0.78L_1$ under these conditions; this estimate was made using the flow-rate amplitude at the acoustic network (Eq. (S11)), so it is only an approximation.

Figure 8b considers the same measurements but with decreased main impedance, increasing $\dot{E}_{2,n1}$ and $|U_1|$ to almost double their cold-impedance values (Fig. S14). For this configuration, the loss seems to onset at $f < 3.4$ Hz. This qualitatively agrees with Eq. (4), which suggests that−with higher $|U_1|$−the same critical $|\xi_1|$ should be reached at higher $f$. The bottom row of Fig. 8 confirms that loss onsets at roughly the same $2|\xi_{1,1}|/L_1$ for the two needle settings. The magnitude of the loss and its impact on cooling power can be tremendous. Compare the data at 1.4 and 3.4 Hz: even though $\dot{E}_{2,n1}$ is about 140 W higher at 1.4 Hz, $\dot{Q}_{c,1}$ is almost 190 W lower. This is likely explained by $|\xi_{1,1}|$ being nearly

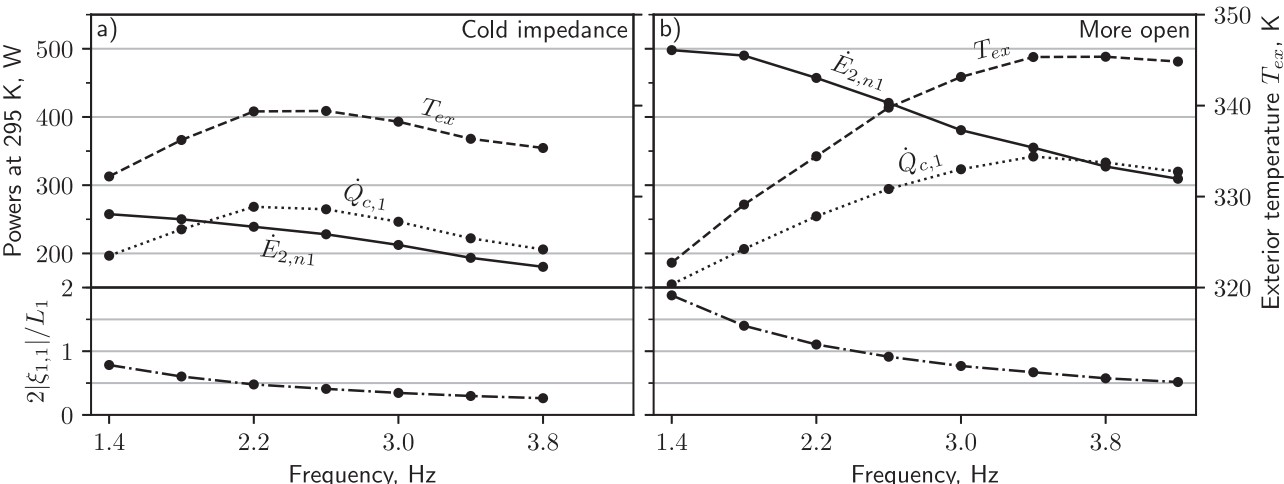

**Fig. 8 | Cooling power loss at the first stage.** First-stage cooling power and acoustic power at the terminating network inlet as a function of $f$. Subplot (**a**) is for cold impedance and (**b**) is for decreased impedance (more open). At some frequencies $\dot{E}_{2,n1}$ is less than $\dot{Q}_{c,1}$, so a significant amount of acoustic power may be dissipated in the first-stage heat exchangers or at the adiabatic-to-isothermal

transition at the ends of the buffer tube (making $\dot{E}_{2,bt} > \dot{E}_{2,n}$); this same observation was not made at the second stage (Fig. S13). Also plotted is the exterior temperature $T_{ex}$ of the PTR. The bottom row shows the approximate stroke of the helium in the first-stage buffer tube normalized by the buffer tube length.

equal to $L_1$—almost all helium in the buffer tube thermally shorts the cold and warm heat exchangers each cycle.

The temperature of the ambient part of the PTR also supports our large-stroke hypothesis. We mounted a thermometer to the exterior of the aluminum block that contains the warm-end heat exchangers and needle valves; this temperature $T_{ex}$ follows the trend of $\dot{Q}_{c,1}$ (Fig. 8). When large-stroke losses are dominant $T_{ex}$ is lowest, even though more acoustic power is being dissipated in the main needle valves. This suggests that helium in the buffer tube is transporting heat from the ambient part of the PTR to the cold-end.

The discrepancy between $\dot{E}_{2,n1}$ and $\dot{Q}_{c,1}$ at 295 K shows that the first stage of this PTR cannot utilize the full output of the compressor, even in the cold-acoustics configuration. It should be possible to significantly increase $\dot{Q}_{c,1}$ near 295 K by redesigning the first-stage buffer tube to accommodate more acoustic power. Large-stroke losses appear to be less significant at the second stage in the cold-acoustics configuration (Fig. S13). This results in $\dot{Q}_{c,2}$ being nearly fixed at 195 K and above (where $\dot{E}_{2,n}$ for both stages change little, Fig. S15), while $\dot{Q}_{c,1}$ drops significantly as $T_w - T_c$ increases (Fig. S3).

We hypothesize that large-stroke losses explain the discrepancy between $\dot{Q}_c$ and $\dot{E}_{2,n}$ at low $f$; however, heat transported through the boundary layer inside the buffer tube is another loss mechanism inversely proportional to $f$ (for details consult[24], specifically the non-conductive $dT_m/dx$ term of the total power flow equation). It is challenging to discern between such loss mechanisms without additional measurements.

## Discussion

Efficient coupling between compressors and pulse tube refrigerators requires dynamic acoustic tuning because of temperature-dependent behavior in regenerators and buffer tubes. Today's commercial PTRs are tuned only for operation at their base temperature, leading to poor cooldown speed and wasted compressor power input.

We developed simple analytics to guide the optimization of PTRs as a function of temperature; acoustic power measurements confirm that these analytics are qualitatively accurate. However, cooling power measurements suggest that the buffer tubes of modern PTRs need to be redesigned for the refrigerators to utilize all power their compressors can produce.

Given the ubiquity of PTRs in modern cryogenics, the improvements to cooldown speed demonstrated in this initial work (1.7 to 3.5 times the original) show that dynamic acoustic optimization can have a widespread impact. Our methods can be readily incorporated into commercial PTRs; however, heat exchange between stages is also required to achieve the greatest improvements, as the cooling power of both refrigeration stages is significantly increased with dynamic acoustic optimization. In experiments with massive payloads such as CUORE, our methods can save a week or more of time each cooldown. In smaller experiments for prototyping quantum circuits where cooldown times are presently comparable to characterization times, dynamic acoustic optimization can substantially increase measurement throughput.

Alternatively, experiments that achieve acceptable cooldown times with oversized PTRs can instead use smaller, dynamically optimized PTRs and consume far less electricity in steady state. If quantum technologies such as quantum networks with large numbers of nodes are ever to become commonplace, critical components must either work at warmer temperatures or the cryogenics required to reach temperatures of a few kelvin and below must become simpler and more efficient. Dynamic acoustic optimization is an important step towards realizing such cryogenics.

The methods shown in this work are not applicable to high-frequency PTRs driven by linear compressors. Furthermore, this study addresses only one of the contemporary issues regarding coupling between low-frequency PTRs and their compressors. For example, the rotary valve that alternatively connects the PTR to low and high-pressure sides of the compressor introduces multiple losses, including leakage and the destruction of work availability from free expansion[17,25]. The methods described here may increase the rotary valve losses at temperatures above base because the portion of the flow that was shunted through the relief valve in the compressor is now sent through the rotary valve and utilized in the PTR. After the cooldown is complete, dynamic acoustic optimization has no impact on efficiency because all acoustic parameters are restored to those that optimize performance at base temperature.

## Methods

The general methodology of this study is given in previous sections. Some topics require additional details, which are included in the Supplementary Information. These details include the derivation of Eqs. (1) and (2), equations for calculating acoustic power from pressure measurements, information on the construction and performance of the thermosiphons, and methods for the emulation of larger loads in the second stage.

## Data availability

The conclusions of this study are supported by the data presented in the main manuscript and Supplementary Information. The data presented in the main manuscript are available in the Source Data file. Source data are provided with this paper.

## Code availability

No specific code or software was required to process the data collected during this study.

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

## Acknowledgements

We acknowledge support from the Professional Research Experience Program between the University of Colorado Boulder and NIST under award number 70NANB18H006. Manuscript is a contribution of NIST, and not subject to copyright. Certain equipment, instruments, or materials are identified in this paper in order to specify the experimental procedure adequately. Such identification is not intended to imply recommendation or endorsement by NIST, nor is it intended to imply that the materials or equipment identified are necessarily the best available for the purpose. We thank Gregory Swift and Ray Radebaugh for their helpful comments when reviewing this manuscript.

## Author contributions

R.S.: conceptualization, methodology, investigation, data curation, writing, and visualization. V.K.: conceptualization, methodology, and supervision. S.B.: conceptualization, methodology, and supervision. J.U.: supervision, funding acquisition, and writing.

## Competing interests

R.S., V.K., and S.B. have submitted a patent through the University of Colorado Boulder on the dynamic acoustic tuning of cryocoolers. The patent has been submitted with International Patent Application No. PCT/US2023/014047. The authors declare no other competing interests.
