## [Peer Review File · Nature Communications]

Dynamic acoustic optimization of pulse tube refrigerators for rapid cooldownREVIEWER COMMENTS

Reviewer #1 (Remarks to the Author):

This manuscript introduces a technological adaptation to a low frequency orifice type pulse tube refrigerator. A lumped model of the system is introduced and the impact of hardware changes on the cooling power is described. The authors claim to have made significant progress in the operation of these type of cryocoolers. The scientific finding, namely, tuning the acoustic impedance to increase the cooldown speed is not novel and is reported earlier (Radebaugh, R., O'Gallagher, A., Lewis, M.A. and Bradley, P.E., 2007. Proposed rapid cooldown technique for pulse tube cryocoolers. International Cryocooler Conference.). You will find more publications on this topic that can be retrieved by following citations to the above-mentioned paper.

Furthermore, as it is well known in the cryocooler community, inertance type pulse tube cryocoolers are more efficient compared to an orifice type cryocooler because the phase of massflow with respect to the pressure in an inertance type is more conducive at the cold heat exchanger compared to an orifice type. Although the authors work is mainly focused on the compressor side, it is not clear why they chose an orifice type pulse tube cryocooler to validate their work. Please explain.

The authors should put in more effort to describe their work in the broader context of existing literature in the fast cooldown.

In view of a lack of novelty, I do not recommend publication of this work in Nature Communications Journal.

Reviewer #2 (Remarks to the Author):

The submitted manuscript represents a solid and impressive piece of work. It provides a combination of applied acoustic theory, experimental confirmation, and helpful supplemental material and presents a compelling approach to improve the performance of pulse tube cryocoolers.

Those of us who have used pulse tube cryocoolers to cool large scale equipment down to liquid helium temperatures will recognize the significance of this work and appreciate the method for reducing the cooldown time. More importantly, with the anticipated scale-up in sub-kelvin cooling for quantum computing, the authors' claim regarding the importance of the findings is legitimate. Nevertheless, it will strengthen the authors' claim if they provide some quantitative, even order of magnitude, estimate of the potential market growth. In fact, the authors do not cite a significant publication related to their topic of study:

Radebaugh, R., et al. Proposed rapid cooldown technique for pulse tube cryocoolers. In Cryocoolers. 2007: ICC Press, Boulder, CO.

However, at the time of the earlier publication, the anticipated growth of the sub-kelvin cooling did not exist. Certainly, the present authors' contribution is timely.

Regarding the earlier publication, both the submitted manuscript and the prior publication utilize adjustments to the acoustic features of the OPT system to decrease the cooldown time by a factor of two or more. The authors of the present manuscript need to assess the earlier publication, compare the approach and findings of the two documents, and identify the new and unique contributions of their work considering the prior publication. The authors' contribution is sufficiently different and with a possibility of automating the acoustic adjustments during cooldown, presents a clear advantage over the prior work. Nevertheless, the prior work forms a significant benchmark and should be discussed.

A few additional detailed comments are as follows:

1. A significant thermodynamic loss for the low frequency pulse tube occurs across the inlet and exhaust valves where the large and time dependent pressure drop is unavoidable. The authors should comment on the degree to which reducing the acoustic resistance (changing from constant pressure to constant flow source) will impact the acoustic loss across these valves.
2. Footnote 6 on page 3: The authors claim that "The analysis of this section is only quantitatively accurate for sine waves, but is qualitatively useful for all waveforms between square and sinusoidal" is not obvious. Some discussion regarding the extension of thermo-acoustic theory based on sinusoidal oscillations to that of periodic square waves would be of value.
3. Equations 1 and 2 are derived from thermo-acoustic theory. A reference for their specific derivation should be provided. Alternately, their derivation could be included in the supplementary information.
4. Page 6, lines 409-412: Clarify the meaning of S2 and S1. Are these the sensible heats of the thermosiphon fluids over the temperature range? How is the ratio adjusted between values of 2 and 9? (See fig. 6b). If the sensible heat is defined by integrating heat capacities of copper over the temperature range of 4 K to 295 K, they should be named enthalpy values. "Sensible heat" refers to the integral of a fluid's heat capacity outside the range of a phase change. Clarify how the combination of the copper thermal load and that from the thermo-siphon fluid contributes to the overall values of S2 and S1.
5. Page 6, lines 423-430. This paragraph is confusing. Is the left-most data point gathered without the use of the thermosiphons? "... without heat transfer between the stages ..." If so, state that fact.

Reviewer #3 (Remarks to the Author):

The manuscript focuses on improving the efficiency of pulse tube refrigerators (PTRs). The author pointed out that traditional PTRs only optimize operation at the base temperature, and also focused on solving the inefficiency problem of PTRs during the process of cooling from room temperature to the base temperature. The manuscript proposes dynamic acoustic optimization of a commercial PTR and shows that the cooldown speed can be increased to 1.7 to 3.5 times the original value. This dynamic acoustic optimization involves adjusting the acoustic network based on temperature, thereby improving performance and reducing energy consumption. This discovery has important implications for research areas that require cryogenic temperatures, such as quantum computing, improving the accessibility of PTRs and reducing operating costs.

The following points should be modified:

1. Writing of this manuscript needs to be improved, for example, "PTRs (Pulse tube refrigerators)" is widely used in the manuscript, but "buffer tube" is used instead of "Pulse tube" when going to the pulse tube alone. And the description of the manuscript is too complicated. For example, the content of the impedance and acoustic power analysis of the refrigerator in the "Introduction" is repeated with the content in the Result. Please simplify similar repetitive content in the manuscript.
2. Figure 1 needs to be improved; it gives mixed information. First of all, the two diagrams do not distinguish between a and b, and some parts are even interlaced together. Secondly, the positions marked by the "Terminating RC net" in the diagram above will make it difficult to connect the two reservoirs and needle valves in the diagram above with the "Terminating RC network" in the figure below. Finally, the words "network" and "regenerator" are used both in full and in abbreviations in the same figure. This also makes the statements of Lines 120 to 123 difficult to be understood directly. Please start by introducing the components shown in Figure 1.
3. "Nomenclature" should be added before the "Introduction".
4. In lines 141 to 142, "E₂ generated in the same PTR can easily be doubled by decreasing R in the terminating networks." cannot be proved, at least there are no formulas before lines 141-142 that could support this conclusion. Refer to comment 1, please carefully adjust the structure of the manuscript.
5. Figures S1-S16 should be placed in the manuscript, because when the data shown in these figures is mentioned in the manuscript, it is necessary to promptly guide the position of the

figures. While making this modification, please check whether the number of figures in the manuscript meets the requirements of the journal.

6. In Figure 2, "The compressor-motor system functions as a pressure or flow rate source, depending on whether the relief valve is open or closed.". Please explain the role of the relief valve in the refrigeration system cause under normal circumstances, the relief valve only prevents the cold head pressure from being too high and plays the role of pressure relief.

7. In Figure 2, the pulse and the regenerator should be connected in series. But there is no pulse tube in the Figure 2, if the volume of pulse tube could be ignored without affecting the results. Please add a schematic diagram of the structure of PTR, and correspond it to the resistance, capacitance, voltage and other parameters in the electrical circuit diagram to make it easier to understand.

8. In the "Cooldown speed comparison", it is noted that the refrigeration performance of the second-stage cold end of the refrigerator at low temperature seems not to be revealed, and the introduction of thermosiphon can significantly improve the cooling speed.

9. In the dynamic-acoustic optimization section, only qualitative optimization analysis and suggestions was proposed, which has limited guiding significance for the industrial optimization of pulse tube refrigerators. In the dynamic and acoustic optimization part, more quantitative analyzes and conclusions should be put forward based on the pulse tube refrigerator modeled in this manuscript to further enhance the research significance of this manuscript.

This study requires significant improvement and clarification before it can be published in the journal.

Response to reviewers: Dynamic acoustic optimization of pulse tube refrigerators for rapid cooldown

We thank all the reviewers for their valuable feedback. In this document, the original reviewer remarks are in black, and our responses are in red.

We believe that many in the cryogenic community will find our work important. Reviewer #2 says, “Those of us who have used pulse tube cryocoolers to cool large scale equipment down to liquid helium temperatures will recognize the significance of this work and appreciate the method for reducing the cooldown time.” and reviewer #3 says, “This discovery has important implications for research areas that require cryogenic temperatures, such as quantum computing, improving the accessibility of PTRs and reducing operating costs.”

Reviewer #1 (Remarks to the Author):

This manuscript introduces a technological adaptation to a low frequency orifice type pulse tube refrigerator. A lumped model of the system is introduced and the impact of hardware changes on the cooling power is described. The authors claim to have made significant progress in the operation of these type of cryocoolers. The scientific finding, namely, tuning the acoustic impedance to increase the cooldown speed is not novel and is reported earlier (Radebaugh, R., O'Gallagher, A., Lewis, M.A. and Bradley, P.E., 2007. Proposed rapid cooldown technique for pulse tube cryocoolers. International Cryocooler Conference.). You will find more publications on this topic that can be retrieved by following citations to the above-mentioned paper.

Furthermore, as it is well known in the cryocooler community, inertance type pulse tube cryocoolers are more efficient compared to an orifice type cryocooler because the phase of massflow with respect to the pressure in an inertance type is more conducive at the cold heat exchanger compared to an orifice type. Although the authors work is mainly focused on the compressor side, it is not clear why they chose an orifice type pulse tube cryocooler to validate their work. Please explain.

The authors should put in more effort to describe their work in the broader context of existing literature in the fast cooldown.

In view of a lack of novelty, I do not recommend publication of this work in Nature Communications Journal.

We thank the reviewer for calling attention to the Radebaugh paper from 2007 concerning the cooldown of high-frequency PTRs using inertance tubes. We originally did not reference this work because this class of PTR is very different from the type we studied (low-frequency, Gifford-McMahon type). However, we agree with the reviewer that this comparison should be made. We have added two paragraphs to the introduction that discuss previous research on the rapid cooldown of pulse tube refrigerators.

In response to the reviewer's comment, “...it is not clear why they chose an orifice type pulse tube cryocooler to validate their work. Please explain.” we provide the following context:

We study low-frequency PTRs because this is the type that is used to cool large experiments to 4 K and below (commercial dilution refrigerators and adiabatic demagnetization refrigerators are usually precooled by these PTRs). This type of PTR is indeed less efficient than the high-frequency type, because the high-frequency type directly connects the coldhead to the compressor, which operates at the resonant frequency of the system (the coldhead and compressor operate at the same frequency without a rotary motor between them). However, the great benefit of the low-frequency coolers is that they easily reach much colder temperatures than the high-frequency coolers and have high cooling powers at low temperatures—an advantage gained through lower regenerator losses. For example, an off-the-shelf PTR like the PT450 from Cryomech/Bluefors provides 5 W of cooling at 4.2 K for a price competitive to the typical academic laboratory. Contrast this to the state-of-the-art high-frequency PTR aboard the James Webb Space Telescope: this PTR only cools to about 18 K and was developed over many years by the top aerospace cryocooler companies in the United States (for more information, consult the work of the Advanced Cryocooler Technology Development Program, funded by NASA). Because high-frequency aerospace PTRs cannot typically reach temperatures near 4 K themselves, Joule-Thompson cryocoolers must be used in conjunction (there are some exceptions, although they often require helium-3). As we reference in the introduction (refs. 1-7), there are a wide variety of applications that presently depend on low-frequency coolers.

The rapid cooldown methods of the Radebaugh paper from 2007 are mostly applicable to high-frequency coolers. Radebaugh shows that for a high-frequency PTR with RLC terminating network, it is important to change the size of the reservoir during cooldown. This is in contrast to our work, where we show that it is critical to change the impedance of the main orifice and the frequency of the rotary valve. Radebaugh concludes that his type of refrigerator cannot benefit from dynamic frequency changes. Radebaugh also says, “The proposed method makes use of the resonance phenomenon that occurs with an appropriately sized combination of inertance tube and reservoir volume”. The difference in acoustic optimization strategies is rooted in the fundamentally different way low-frequency and high-frequency PTRs are driven: low-frequency PTRs use scroll compressors that are coupled to the refrigerator using rotary valves, while high-frequency PTRs use linear compressors that are directly connected to the refrigerator, and inertance tubes are used in the acoustic network (or displacers if the PTR is a Stirling type).

There are additional differences between our work and Radebaugh’s:

1. Radebaugh does not present any experimental results. Some experiments were later presented in another work by Lewis, Radebaugh, and others (“Pulse Tube Cryocooler for Rapid Cooldown of a Superconducting Magnet”), which we also reference in the manuscript. However, Lewis et al. does not provide data that compares overall cooldown speed, they only show marginal improvement to cooling power as a function of temperature.
2. It is not possible to shunt flow in a high-frequency cooler. The flow that is shunted through the relief valve in low-frequency PTR compressors is one of the fundamental reasons why dynamic optimization is needed and is one of the reasons why such large changes to cooldown speed were demonstrated by us (up to 3.5 times the original cooldown speed). If flow is shunted then the compressor operates as a pressure source (and acoustic power is determined by our Equation 1), while if no flow is shunted then the compressor operates as a flow source (our Equation 2).

In the main manuscript we also now bring to attention a work by Kim et al. (“Research on fast cool-down of orifice pulse tube refrigerator by controlling orifice valve opening”) that attempted to study cooldown speed of a low-frequency cooler. However, this work is insufficient in multiple ways:

1. They do not optimize frequency, but only study a few changes to the main impedance.
2. They only study cooldown to 100 K. This is important because dynamic acoustic optimization is needed more the larger the overall temperature span. The attenuation of flow through the regenerators is proportional to the ratio of densities at the cold-end temperature and warm-end temperature. This ratio is only 3 for Kim et al.'s single-stage refrigerator but is almost 80 for the two-stage commercial refrigerator we study.
3. They achieve a cooldown speed improvement of less than 10%. This is likely related to the above point: dynamic acoustic optimization is not as much needed for a cold-end temperature of 100 K.
4. They study a custom PTR, not a commercial one. Results from a commercial PTR are more translatable to the community of cryogenic users.
5. They do not provide any analysis that motivates how/why to perform dynamic acoustic optimization, like we do (equations 1-3).

We also include the below table, which summarizes our work compared to the Radebaugh, Lewis, Kim, and Yan publications. We do not discuss the Yan work (“Investigation of improving cool-down speed of Stirling type pulse tube cryocooler with ambient displacers”) in detail in this response because (like the Radebaugh/Lewis works) they study a high-frequency PTR driven by a linear compressor. However, we do reference the Yan publication in the manuscript’s introduction.

Work	This manuscript	Radebaugh (2007) / Lewis (2009)	Kim et al. (2010)	Yan et al. (2020)
Operating frequency	1.4 Hz to 4.2 Hz	60 Hz (static)	4 Hz (static)	59 Hz to 70 Hz
Pulse tube type	2-stage, dual-inlet, RC terminating network	1-stage, RLC terminating network	1-stage, RC terminating network	1-stage Stirling (ambient displacer)
Compressor type	Oil lubricated scroll plus rotary motor	Resonant linear	Likely an oil lubricated scroll plus rotary motor	Resonant linear
Temperature span	295 K – 3 K	300 K – 67 K	295 K – 100 K	300 K – 77 K
Demonstrated cooldown speed (normalized by default speed)	1.7 to 3.5, depending on cryostat configuration (experimental)	Radebaugh gives no experimental results but calculates the speed as 1.94; Lewis shows marginally increased cooling powers but does not measure cooldown speed	1.09 (experimental)	1.1 (simulation)
Dynamic acoustic changes	Frequency, main orifice impedance	Reservoir size	Main orifice impedance	Frequency, compressor voltage
Analytic result for how/why dynamic optimization?	Yes (equations 1-3)	No	No	No
Commercial PTR studied?	Yes	No	No	Yes
Study of relief valve?	Yes	No	No	No
Applicable to high-freq. PTR?	No	Yes	No	Yes
Applicable to low-freq. PTR?	Yes	No	Yes	No

With the previous points in mind, we believe that our work is sufficiently novel. We believe that cryogenic users who depend on cooling to 4 K and below (a large community) will find our work valuable

because they can cool their experiments significantly more quickly and/or use PTRs that more efficiently use the acoustic power that their compressors generate. This goal was not achieved previously because the existing literature studied different classes of PTRs, studied PTRs operating at significantly higher cold-end temperatures, and did not demonstrate significant enhancements to cooldown speed (as we do).

Reviewer #2 (Remarks to the Author):

The submitted manuscript represents a solid and impressive piece of work. It provides a combination of applied acoustic theory, experimental confirmation, and helpful supplemental material and presents a compelling approach to improve the performance of pulse tube cryocoolers.

Those of us who have used pulse tube cryocoolers to cool large scale equipment down to liquid helium temperatures will recognize the significance of this work and appreciate the method for reducing the cooldown time. More importantly, with the anticipated scale-up in sub-kelvin cooling for quantum computing, the authors' claim regarding the importance of the findings is legitimate. Nevertheless, it will strengthen the authors' claim if they provide some quantitative, even order of magnitude, estimate of the potential market growth. In fact, the authors do not cite a significant publication related to their topic of study:

We attempted to contact PTR manufacturers about the size of the market and therefore the impact of our methods to the 4 K and below cryogenics community. However, this information is not public, and we are therefore unable to provide an estimate.

Radebaugh, R., et al. Proposed rapid cooldown technique for pulse tube cryocoolers. In Cryocoolers. 2007: ICC Press, Boulder, CO.

However, at the time of the earlier publication, the anticipated growth of the sub-kelvin cooling did not exist. Certainly, the present authors' contribution is timely.

Regarding the earlier publication, both the submitted manuscript and the prior publication utilize adjustments to the acoustic features of the OPT system to decrease the cooldown time by a factor of two or more. The authors of the present manuscript need to assess the earlier publication, compare the approach and findings of the two documents, and identify the new and unique contributions of their work considering the prior publication. The authors' contribution is sufficiently different and with a possibility of automating the acoustic adjustments during cooldown, presents a clear advantage over the prior work. Nevertheless, the prior work forms a significant benchmark and should be discussed.

We thank the reviewer for this feedback concerning Radebaugh's work. Please see our response to reviewer #1 where we make extensive comments about the difference between low-frequency and high-frequency PTRs. We have also added two paragraphs to the introduction discussing previous literature on the rapid cooldown of PTRs.

A few additional detailed comments are as follows:

1. A significant thermodynamic loss for the low frequency pulse tube occurs across the inlet and exhaust valves where the large and time dependent pressure drop is unavoidable. The authors should comment

on the degree to which reducing the acoustic resistance (changing from constant pressure to constant flow source) will impact the acoustic loss across these valves.

We thank the reviewer for this important point. We have added a paragraph to the conclusion discussing this topic.

2. Footnote 6 on page 3: The authors claim that “The analysis of this section is only quantitatively accurate for sine waves, but is qualitatively useful for all waveforms between square and sinusoidal” is not obvious. Some discussion regarding the extension of thermo-acoustic theory based on sinusoidal oscillations to that of periodic square waves would be of value.

We have added a section (Supplementary Methods A) which compares the result between square and sinusoidal waves.

3. Equations 1 and 2 are derived from thermo-acoustic theory. A reference for their specific derivation should be provided. Alternately, their derivation could be included in the supplementary information.

This derivation is now given in Supplementary Methods B.

4. Page 6, lines 409-412: Clarify the meaning of S2 and S1. Are these the sensible heats of the thermosiphon fluids over the temperature range? How is the ratio adjusted between values of 2 and 9? (See fig. 6b). If the sensible heat is defined by integrating heat capacities of copper over the temperature range of 4 K to 295 K, they should be named enthalpy values. “Sensible heat” refers to the integral of a fluid’s heat capacity outside the range of a phase change. Clarify how the combination of the copper thermal load and that from the thermo-siphon fluid contributes to the overall values of S2 and S1.

We changed the vocabulary from “sensible heat” to “enthalpy difference” throughout the manuscript. This was calculated only by considering the change in enthalpy of the solids, not the fluids in the thermosiphons. We have added new information in Supplementary Methods D, which tells how we calculated the enthalpy difference and why the enthalpy of the fluids was ignored (it is very small compared to the enthalpy of the solids). The ratio is adjusted by changing the virtual mass applied to the second stage, as stated in the following sentence of the manuscript: “To study the effect of $\Delta h_2/\Delta h_1$, we varied the virtual mass on the second stage and measured the rapid cooldown speed...”

5. Page 6, lines 423-430. This paragraph is confusing. Is the left-most data point gathered without the use of the thermosiphons? “... without heat transfer between the stages ...” If so, state that fact.

Thank you for the comment. We have reworked this paragraph stating that we removed the copper load, which also disabled the function of the thermosiphons.

Reviewer #3 (Remarks to the Author):

The manuscript focuses on improving the efficiency of pulse tube refrigerators (PTRs). The author pointed out that traditional PTRs only optimize operation at the base temperature, and also focused on solving the inefficiency problem of PTRs during the process of cooling from room temperature to the base temperature. The manuscript proposes dynamic acoustic optimization of a commercial PTR and shows that the cooldown speed can be increased to 1.7 to 3.5 times the original value. This dynamic acoustic optimization involves adjusting the acoustic network based on temperature, thereby improving performance and reducing energy consumption. This discovery has important implications for research areas that require cryogenic temperatures, such as quantum computing, improving the accessibility of PTRs and reducing operating costs.

The following points should be modified:

1. Writing of this manuscript needs to be improved, for example, “PTRs (Pulse tube refrigerators)” is widely used in the manuscript, but “buffer tube” is used instead of “Pulse tube” when going to the pulse tube alone. And the description of the manuscript is too complicated. For example, the content of the impedance and acoustic power analysis of the refrigerator in the “Introduction” is repeated with the content in the Result. Please simplify similar repetitive content in the manuscript.

We thank the reviewer for this comment. We have added the following footnote (#3) the first time we use the term buffer tube:

“Also called the pulse tube. We prefer to use the term thermal buffer tube or buffer tube for short because it is unambiguous that we are referring to a component of the pulse tube refrigerator and not the refrigerator itself.”

To simplify some instances of repetitive content between the introduction and results section we removed some discussion of acoustic power in the third-to-last and fourth-to-last paragraphs of the introduction, as well as a line about the pressure relief valve in the third paragraph of the first results section. We also removed a repetitive definition in the Fig. 8 caption. Please see the highlighted text in the revised manuscript for all changes.

2. Figure 1 needs to be improved; it gives mixed information. First of all, the two diagrams do not distinguish between a and b, and some parts are even interlaced together. Secondly, the positions marked by the “Terminating RC net” in the diagram above will make it difficult to connect the two reservoirs and needle valves in the diagram above with the “Terminating RC network” in the figure below. Finally, the words “network” and “regenerator” are used both in full and in abbreviations in the same figure. This also makes the statements of Lines 120 to 123 difficult to be understood directly. Please start by introducing the components shown in Figure 1.

We thank the reviewer for these suggestions. We have made the following changes to Figure 1:

1. Added a) and b) labels.
2. We relabeled the terminating RC networks on both the schematic and photograph.
3. We removed any abbreviations from the figure.
4. The relief valve is now shown in the schematic.

The most important components of Fig. 1 are introduced in the second paragraph of the introduction, where Fig. 1 is first referenced. The other components of Figure 1 are explained later in the introduction once discussion of the RC network becomes relevant.

3. “Nomenclature” should be added before the “Introduction”.

A nomenclature section is not allowed by the journal.

4. In lines 141 to 142, “ E_2 generated in the same PTR can easily be doubled by decreasing R in the terminating networks.” cannot be proved, at least there are no formulas before lines 141-142 that could support this conclusion. Refer to comment 1, please carefully adjust the structure of the manuscript.

Thank you for the comment. We have removed this statement from the introduction. The reader can still see this result in Fig. 8, which shows a doubling of acoustic power at 1.4 Hz.

5. Figures S1-S16 should be placed in the manuscript, because when the data shown in these figures is mentioned in the manuscript, it is necessary to promptly guide the position of the figures. While making this modification, please check whether the number of figures in the manuscript meets the requirements of the journal.

The maximum number of figures the journal allows is 10, so it is not possible to move all the figures to the main manuscript. We would like our work to be published in Nature Communications so that the

paper is accessible to a wide audience, i.e. both cryogenic experts and cryogenic users who are not experts in the function of PTRs. To make the paper accessible to cryogenics users we made the decision to only include the most essential figures in the main manuscript, which also helps keep the main manuscript succinct. However, as a way to include additional fundamental measurements in the main manuscript that were previously only in the supplementary information, we have added a new Figure 3 which includes pressure measurements at 295 K and 3 K. This data shows that the relief valve is open at the start of cooldown for a cold-optimized PTR.

6. In Figure 2, “The compressor-motor system functions as a pressure or flow rate source, depending on whether the relief valve is open or closed.” Please explain the role of the relief valve in the refrigeration system cause under normal circumstances, the relief valve only prevents the cold head pressure from being too high and plays the role of pressure relief.

Thank you for the comment. You are correct, this is the purpose of the pressure relief valve. We have added this information in the introduction the first time the relief valve is introduced: “typically shunts (wastes) a significant amount of its flow through a pressure relief valve because the impedances of the main needle valves are too high. The pressure relief valve is installed in low-frequency PTR compressors to prevent overpressurization of the refrigerator before it cools; it essentially enables the refrigerator and compressor to be statically (although inefficiently) coupled as pressure and flow rate magnitudes change during cooldown.”

7. In Figure 2, the pulse and the regenerator should be connected in series. But there is no pulse tube in the Figure 2, if the volume of pulse tube could be ignored without affecting the results. Please add a schematic diagram of the structure of PTR, and correspond it to the resistance, capacitance, voltage and other parameters in the electrical circuit diagram to make it easier to understand.

We thank the reviewer for this suggestion. Instead of re-drawing the schematic of a PTR in Fig. 2, we believe it is more succinct to add the following sentence to the Fig. 2 caption: “The schematic of this refrigerator is the same as shown in Fig. 1a except without the second stage and with no bypass.” The volume of the buffer tube is lumped into C_b (as we write, “The other compliance C_b is a lumped parameter accounting for the compressible volume between motor and terminating network, including the buffer tube...”). While the regenerator and buffer tube are in series physically, the compliance from the buffer tube should be in parallel for the circuit (the compressibility of the buffer tube changes the flow rate phasor but not the pressure phasor, see *Thermoacoustics* by Gregory Swift, Fig. 4.11). Although the compliance of the regenerator, helium hose, and buffer tube could be accounted for individually, it would then be unfeasible to analytically solve the circuit. By lumping these components together, we capture the essential physics without overcomplicating the circuit.

8. In the “Cooldown speed comparison”, it is noted that the refrigeration performance of the second-stage cold end of the refrigerator at low temperature seems not to be revealed, and the introduction of thermosiphon can significantly improve the cooling speed.

Thank you for the comment. The cooling power near 4 K is not affected by dynamic acoustic optimization. This is stated at the end of the “cooldown speed comparison” section: “At that point it is optimal for the PTR to revert to cold acoustics.”, where cold acoustics are the settings that are traditionally used to optimize PTRs. Furthermore, in the last paragraph of the conclusion we have added the following line: “After cooldown is complete dynamic acoustic optimization has no impact to efficiency because all acoustic parameters are restored to those that optimize performance at base temperature.”

We also quantify how much cooldown speed can be improved without the use of thermosiphons/heat transfer by the data point in Figure 7 labeled “optimized acoustics”, in contrast to the line labeled

“Optimized acoustics and heat transfer”. The cooldown speed without thermosiphons but with acoustic optimization is 1.7 times the default cooldown speed, which is still a very significant improvement. We would also like to note that the first stage cooling power can be improved by almost 150 W at the first stage with acoustic optimization at 295 K. This represents almost a 3x improvement over the default cooling power at the second stage. While a heat transfer mechanism is required to realize this gain in cooling power at the second stage, the source of the cooling power increase is still from dynamic acoustic optimization.

9. In the dynamic-acoustic optimization section, only qualitative optimization analysis and suggestions was proposed, which has limited guiding significance for the industrial optimization of pulse tube refrigerators. In the dynamic and acoustic optimization part, more quantitative analyzes and conclusions should be put forward based on the pulse tube refrigerator modeled in this manuscript to further enhance the research significance of this manuscript.

We thank the reviewer for this comment. Quantitative advice on how to optimize a generic commercial PTR is extremely challenging because each model of PTR is constructed differently. For example, Bluefors/Cryomech sells 8 different models and sizes of 4 K, low-frequency PTRs, and Sumitomo also sells many models. While we do give the quantitative changes to frequency and needle position as a function of temperature in supplementary Figures S10 and S11, these details will not be directly usable for any PTR except the particular model used in our study (Cryomech PT407). Those details are not directly translatable to other PTRs that have different needle valves or reservoir sizes.

We believe that the best advice to manufacturers and users of PTRs is to explain the fundamental physics (Equations 1, 2, and 3) and the general methodology, which we discuss at the end of the “Basis for temperature dependent optimization” section: “the optimal main impedance is smallest at ambient temperature and increases towards the cold-impedance value as the temperature drops” and “The optimal f is highest at ambient temperature and decreases as temperature drops.”

We would also like to note that we make a significant practical observation in the last results section titled, “Loss mechanisms”. In that section, our measurements show that manufacturers must redesign their buffer tubes to take full advantage of all the acoustic power these compressors can generate when the refrigerator is warm.

This study requires significant improvement and clarification before it can be published in the journal.

REVIEWERS' COMMENTS

Reviewer #2 (Remarks to the Author):

The authors have addressed each of the concerns expressed in my initial review. I consider the manuscript ready for publication. Furthermore, based on their comparison with prior publications regarding decreased cooldown of PTRs, I also consider the work novel and of significant relevance for the anticipated sub-kelvin cooler applications.

Reviewer #3 (Remarks to the Author):

Thank you for your reply to our review and the revision of the manuscript. There is no further question and comment.